# Midwifery centers as enabled environments for midwifery: A quasi experimental design assessing women's birth experiences in three models of care in Bangladesh, before and during covid

**Jennifer Rebecca Stevens**[1]*, **Lora L. Sabin**[1], **Monica A. Onyango**[1], **Malabika Sarker**[2], **Eugene Declercq**[1]

**1** School of Public Health, Boston University, Boston, MA, United States of America, **2** James P Grant School of Public Health, BRAC University, Dhaka, Bangladesh

\* jencnm@gmail.com

## Abstract

### Background

The midwifery model of care is a human rights-based approach (HRBA) that is unique and appropriate for the majority of healthy pregnant women, yet full expression may be limited within the medical model. Midwifery centers are facilities designed specifically to enable the practice of midwifery. In high resource countries, they have been shown to be cost effective, evidence-based, avoid over medicalization, and provide safe, efficient and satisfying care.

### Methods

A quasi-experimental design was used to assess the impact of three models of care on women's experiences of respect, and trust in maternity care provision, both before and during the pandemic in Bangladesh, as well as their fear and knowledge around COVID-19, during the pandemic. The models were: "fully enabled midwifery" ("FEM") in freestanding midwifery centers; "midwifery and medicine" ("MAM") in medical facilities with midwives working alongside nurses and doctors; and "no midwifery" ("NoM") in medical facilities without midwives. Phone survey data were collected and analyzed from all women (n = 1,191) who delivered from Jan 2020-June 2020 at seven health care facilities in Bangladesh. Comparison of means, ANOVA, post hoc Tukey, and effect size were used to explore the differences in outcomes across time periods.

### Findings

Pre-pandemic, women served by the FEM model reported significantly higher rates of trust and respect ($p<0.001$) compared to the NoM model, and significantly higher rates of trust ($p<0.001$) compared to MAM. During the pandemic, in the FEM model, the experiences of respect and trust did not change significantly from the pre-pandemic rates, and were

**Data Availability Statement:** All relevant data are within the manuscript and its Supporting Information files.

**Funding:** The authors received no specific funding for this work.

**Competing interests:** The authors have declared that no competing interests exist.

significantly higher than both the MAM and NoM models (p < 0·001). Additionally, during the pandemic, women served by the FEM model had the lowest experience of COVID fear (p<0·001).

## Interpretation

Fully enabled midwifery in midwifery centers had a significantly positive effect on woman's experience of respect and trust in care compared to the other models, even in the context of a pandemic.

## Introduction

The *Lancet Maternal Health* series in 2006 identified facility-based-birth with a skilled care provider as key to improving maternal health outcomes [1]. Increasing access to care through these interventions has contributed to lowering the maternal mortality ratio (MMR) globally–dropping it 44% from 1990 to 2015 [2, 3]. Currently increasing access continues to be the focus of many maternal and child health interventions in low- and middle-income countries (LMICs) [4, 5].

Since 2015, global reductions in the MMR have slowed [6, 7]. Continued progress to reduce the MMR may require a deeper understanding of the nature of care, including what quality of care entails, and what drives demand for facility based births and skilled attendants. The follow up *Lancet Maternal Health* series in 2016 found that although rates of facility-based care with a skilled care provider have increased, poor quality of care–including disrespect and abuse, and over medicalization–remain major issues [8].

Redefining quality of care to include women's experience of care reflects not just the technical aspect of care, but the *process of how* that care is provided. The process of care provision impacts women's experience, their care seeking behavior and eventually maternal mortality [9]. In WHO's "*Standards for improving quality of maternal and newborn care in health facilities,*" (2016), provision and experience of care are placed beside each other in the quality of care framework, reflecting their equal importance, stating "there is a complex interplay between experience of care and pregnancy outcome" (pg 5) [10].

A human rights-based approach (HRBA) in maternal care is a key to improving the quality and acceptability of care by engaging women as participants in care decisions [11]. Although the United Nations Population Fund (UNFPA) and others [11, 12] have provided guidance on using a HRBA in care, there is a dearth of on-the-ground examples of implementation [6, 13, 14].

The midwifery model of care is a HRBA that is appropriate for the majority of healthy pregnant women. Yet midwives require an enabling environment for the full expression of the midwifery model. Additionally there often many educational pathways to become a midwife, making evaluation challenging.

In this research, the term "professional midwife" will be used to describe midwives who meet the educational standards of the International Confederation of Midwives (ICM), while the term "non-standard midwife" will refer to those working as midwives who do not meet the ICM educational standards [15, 16].

An enabling environment is often described as being safe, having adequate staff, regular pay, and the supplies needed to provide care [17]. In hospital spaces doctors and nurses provide care using the medical model based on a curative and intervention-oriented approach

which is appropriate for ill-health and disease. Midwives, may require additional support and a unique space to provide the relationship-focused midwifery model of care. When working side by side, these different approaches to care, between the midwifery and medical care models, may impact the full implementation of midwifery and a woman's experience of care.

Freestanding midwifery centers are community-based health care facilities that provide an enabling environment for midwifery care. Care is provided exclusively by professional midwives and includes, sexual and reproductive health care, antenatal, intrapartum, and post-partum care [15]. They do not offer cesarean delivery, but maintain referral relationships with hospitals so they can transfer women seamlessly if needed. Although similar to a clinic, the midwifery center is more home-like, reflecting the partnership between woman and midwife in the care process [18]. Midwifery centers in LMIC provide an enabling environment for midwives to provide compassionate normal birth care, improve access, and strengthen networks of care [19–21]. Additionally, freestanding midwifery centers can free up hospital beds used by healthy birthing women, for urgent care to the acutely ill, offering resilience to a health system especially in a crisis.

## Aim of the study

In this analysis, a quasi-experimental design was used to answer the question, "what is the impact of three different models of birth care on women's experiences of respect and trust, before and during the pandemic in Bangladesh, as well as on their fear and knowledge around COVID-19, during the pandemic".

The three models identified and named for this research were; (1) fully enabled midwifery (FEM), (2) midwifery and medicine (MAM), and (3) no midwifery (NOM). The FEM model examined care provided exclusively by professional midwives in the enabling environment of freestanding midwifery centers. The MAM model examined care provided by professional midwives working in standard midwifery-led care units, alongside doctors, non-standard midwives and nurses in hospitals. The NOM model examined care provided by doctors, non-standard midwives and nurses in hospitals. No professional midwives worked in these facilities.

## COVID-19 and pregnancy care

Natural and human-caused events that lead to humanitarian crises, such as earthquakes, conflicts, and disease outbreaks, disproportionately impact women [22] with documented increased maternal mortality [23] from, among other things, fear and mistrust keeping women from accessing healthcare in facilities [24]. During the COVID crisis, healthy pregnant women had legitimate fears of COVID-19 exposure when the only access to antenatal and birth care was in a hospital filled with COVID-19 patients. The indirect reproductive "fallout" from COVID-19 is expected to result in increased unintended pregnancies, increased GBV [25], increased child marriages [26], and increased maternal mortality throughout the world [27]. The stress COVID added to health care provided a natural experiment to evaluate different healthcare system models and assess how women experience care during a crisis.

## Bangladesh

From 1990 to 2010, the WHO-estimated MMR for Bangladesh dropped from 574 to 194 per 100,000 live births [28, 29], but since 2010 the MMR has plateaued, with a 2016 MMR of 196 [30].

From 2007 to 2016, the share of facility-based births increased from 15% to 50%, with the greatest increase in private facilities [30]. Bangladesh reports a national cesarean rate of 32.7%,

but of the facility-based births, 67% were delivered by cesarean birth, most as planned surgeries in private facilities [31].

Bangladesh has a structured public health care system covering rural and urban areas. It is comprised of district level hospitals, upazilla level health complexes, union level health and family welfare centers, and thousands of community clinics at the union level with the number of facilities designated by the population they serve [32]. This is supplemented by an extensive private hospital system concentrated in urban centers [32]. With government health expenditure at 2.3% of the country's GDP, out of pocket costs remain very high, excluding the majority from health care [32].

Additionally, care quality is a serious concern. A recent 2016 report found that 26% of women in Bangladesh lacked even one antenatal care visit [31], and the quality of the visits is poor and inconsistent–only 39.7% of women receive information on the danger signs of pregnancy, yet 80.2% receive an ultrasound, a billable procedure [30].

Mahumud et al. (2019) found that women in Bangladesh were more likely to birth in a health facility if they were treated with respect [33]. A profound lack of evidenced-based, respectful care throughout Bangladesh has been documented [34].

The COVID-19 pandemic exacerbated these problems. From January 2020 to May 2020 the impact of COVID-19 resulted in a 50% decline in the utilization of maternal, newborn and family planning services, a 31% decrease in ANC visits, and a greater than 50% decrease in facility-based birth [35].

## Methods

### Design

A quasi-experimental design was used to assess the impact of three models of birth care identified for this research–FEM, MAM and NOM–on women's experiences of respectful care and trust, before and during the pandemic in Bangladesh, as well as on their fear and knowledge around COVID-19, during the pandemic. Although these three models have existed since the introduction of midwifery in Bangladesh in 2018, they were identified and named for this research.

### Participants

**Facilities.**   A purposive, non-probability sample was used to identify seven different facilities that met the study's definitions. The facilities were then invited to participate via an email sent to the professional midwives, and non-standard midwives working in labor and delivery at the facilities.

The FEM model was represented by two midwifery centers, the MAM model in two Upazila Health Complexes (UHC) sub-district hospitals, and the NoM model in two UHCs and one Medical College Hospital (MCH).

The FEM midwifery centers enrolled in this study met the global definition, care was provided solely by professional midwives [36, 37].

The first group of professional midwives graduated and began working in Bangladesh in 2018. Prior to that, nurses, non-standardized midwives and doctors, were the primary providers of care during birth in all health facilities [38, 39].

Over 300 UHC in Bangladesh have professional midwives working in midwifery-led care units in the labor wards alongside nurses, non-standardized midwives and doctors. The two participating UHCs representing the MAM model were selected from these UHCs.

There were no midwives posted in the remaining UHCs, nor in any of the District or Medical College Hospitals. The only providers in the labor wards at these facilities were nurses,

non-standardized midwives and doctors. The two UHCs and one MCH representing the NoM model were selected from these. All groups (FEM, MAM and NoM) had both rural and urban facilities represented.

The MAM and NoM facilities were chosen by purposive sampling based on the models of care and access to patient's mobile phone numbers (not all facilities collected mobile numbers). All public healthcare facilities (i.e. UHC, MCH) in Bangladesh follow the same national clinical protocols, staffing, management and supply guidelines. A summary of the facility data for each model of care can be found in S4 File.

**Women.** Eligibility was limited to women who gave birth vaginally between January-June 2020, with care provided by a professional midwife in FEM and MAM, or by a nurse-midwife (non-standard midwife) in NoM, and provided a mobile number. All eligible women were invited to participate and verbally consented via phone. Participation rates were 73·6% from the FEM model, 71·4% (MAM) and 77% (NoM), for an overall response rate of 74·4%. The target population for NOM and MAM models was set at 363 to match the number of women enrolled in the FEM model. During the study period, from the women who delivered vaginally at the participating facilities, 59·4% pre-COVID and 63·5% during COVID had provided a mobile phone number. The details of the response rates of the participants are summarized in S3 File, and descriptions of women participants are summarized in S5 File.

**Data collectors.** Four Bangladeshi data collectors were trained in-person over a four-week period by a researcher experienced in participatory research, facilitation and data collection. All of the data collectors had at least three years of college. All were female, and spoke Bangla as their first language, with limited English. The research assistant was male, Bangladeshi, spoke Bangla as his mother tongue, and was an experienced researcher and English translator.

**Outcomes studied.** The focus of this research was on the woman's experience of four outcomes; (1) experience of respect during care, (2) trust in the healthcare system, (3) experience of COVID-19 fear, (4) and knowledge of COVID-19. A phone-administered questionnaire consisting of 31 closed-ended, and 11 open-ended questions was used. The closed-ended questions came from two validated tools to assess respect and trust, and modified surveys on Ebola to assess fear (see below). The open-ended questions included the date the woman gave birth–to identify the data as pre-pandemic (January-March 2020) or during the pandemic (April-June 2020), and three demographic questions–her parity, education and monthly family income.

To assess respect, the validated "Mothers on Respect index" (MORi) was used [40]. After piloting this survey, two questions were removed from the original–one on sexual orientation and one on health insurance–as both were felt inappropriate for the Bangladeshi context. Trust was assessed using the "Trust in the public healthcare system" survey created and used in India [41].

Fear associated with COVID-19 was assessed by modifying surveys created for Ebola on fear and stigma, focusing on the woman's personal experience and perception [42]. COVID-19 related knowledge was captured using knowledge questions around masks, transmission, prevention and the cause of COVID-19. All data were collected anonymously via the phone by the data collectors. The survey instruments are included as S1 and S2 Files.

## Ethical approvals and inclusivity in global research

Ethical clearance was obtained from the Boston University Medical Campus institutional review board, USA (H-38365), and from the BRAC School of Public Health, Bangladesh (IRB Reference No. 2019-004-ER). Additional information regarding the ethical, cultural, and

scientific considerations specific to inclusivity in global research is included in the Supporting Information, as S8 File.

## Statistical analysis

After verifying that the data were normally distributed, an initial analysis of the outcomes within each model was performed using ANOVA. A TukeyHSD post hoc test was then done to look for differences between the models, as well as an $Eta^2$ ($\eta^2$) effect size for substantive significance, and to assess for Type 1 error. This process was repeated with the data from the pre-pandemic and during-pandemic periods to assess for any changes both within and between the three models of care. Finally, a multiple linear regression model with dummy variables was used to explore any relationships between outcomes, models and covariates (parity, income, education, facility, and data collector).

## Results

### Participants

Women in the FEM model had higher parity, (Table 1) with 82% of women in the FEM model being multiparous, while 54% in MAM and 62% in the NoM were multiparous (Table 1). Although parity and education were significantly different between the models, analysis by linear regression found no relationship between outcomes, models and covariates in the pre-pandemic period. During the pandemic, higher levels of education and income were associated with higher levels of trust, and higher levels of parity and income were associated with higher levels of respect. Estimates of the strength of the relationships between covariates, derived from the coefficient in the linear model, showed these were of a much smaller magnitude than the effect of the model. (see S6 File for details).

The raw data analysis of results found that women who received care in the FEM model reported the highest means both pre- and during the pandemic for trust, and respect, as well as the highest means for COVID knowledge and the lowest for COVID fear during the pandemic (Tables 2 and 3).

**Table 1. Description of participants in the different care models.**

|  | FEM: | MAM: | NoM: | Total study population: | p value |
|---|---|---|---|---|---|
|  | n = 363 | n = 312 | n = 515 | n = 1190 | Total |
|  | pre: 207 | pre: 190 | pre: 304 | Pre: 715 | Pre-COVID |
|  | C19: 156 | C19: 122 | C19: 211 | C19: 475 | COVID |
| **Years of Education** | 6·51 yrs (3·50) | 7·20 yrs (4·21) | 8·82 yrs (4·52) | 7·69 yrs (4·28) | p<0·001* |
| mean (sd), pre-covid/covid | Pre: 6·24 | Pre: 8·1 | Pre: 8·78 | Pre: 7·86 (4·20) | Pre: <0·001* |
|  | C19: 6·86 | C19: 5·79 | C19: 8·89 | C19: 7·40 (4·37) | C19: <0·001* |
| **Monthly Income (in Taka)** | 10,806 (6,367) | 11,402 (5,599) | 11,434 (5,840) | 11,234 (5,946) | p = 0·258 |
| mean (sd) pre-covid/ covid | Pre: 10,213 | Pre: 11,787 | Pre: 11,809 | Pre: 11,363 (5,943) | Pre: 0·004 |
|  | C19: 11,593 | C19: 10,803 | C19: 10,833 | C: 11,049 (5,952) | C19: 0·385 |
| **Parity** | 1·58 (1·15) | 0·814 (0·95) | 1·10 (1·18) | 1·173 (1·15) | p<0·001* |
|  | Pre:1·67 | Pre: 0·72 | Pre: 1·15 | Pre: 1·18 | Pre: <0·001* |
| mean (sd) | C19: 1·47 | C19: 0·97 | C19: 1·04 | C19: 1·16 | C19: <0·001* |
|  | 67 Nulip (18%) | 145 Nulip (46%), | 194 Nulip (38%), | 406 Nulip (34%) |  |
|  | 296 Multip (82%) | 167 Multip (54%) | 321 Multip (62%) | 784 Multip (66%) |  |

Pre–preCovid19; C19–during COVID19 pandemic; FEM: Fully enabled midwifery; MAM: Midwives and medicine; NoM: No midwifery; Nulip: Never had a baby before; Multip: previously had a baby.

**Table 2. Mean results pre-COVID by model.**

| Outcomes: | FEM- | MAM | NoM |
|---|---|---|---|
| | n = 363 | n = 312 | n = 515 |
| | (pre: 207) | (pre: 190) | (pre: 304) |
| **RESPECT** Pre covid mean (sd) | Pre: 57·3 (4·44) | Pre: 56·1 (4·31) | Pre: 52·7 (8·41) |
| **TRUST** Pre covid mean (sd) | Pre: 36·5 (2·71) | Pre: 35.0 (3·05) | Pre: 33·1 (3·80) |

FEM: Fully enabled midwifery; MAM: Midwives and medicine; NoM: No midwifery.

## Respect and trust

An ANOVA analysis demonstrated that the model of care a woman used for birth was significantly related to her experience of respect and trust ($p < 0.001$) during the pre-pandemic and pandemic periods (Tables 5 & 6).

A Tukey HSD pairwise comparison found a woman's experience of trust in the public healthcare system in the pre-pandemic period (Table 5), and her experience of trust and respect during the pandemic (Table 6), was higher in the FEM model compared to the MAM and NoM model ($p < 0.001$). In the pre-pandemic period, the mean for respect in FEM was significantly higher only when compared to the NoM model ($p < 0.001$) (Table 5).

Analysis on changes in mean levels of outcomes in each model, from pre- to during the pandemic period, found trust and respect as relatively stable in the FEM and NoM models (Table 4). The MAM model demonstrated a higher mean for respect and trust when compared to the NoM model in the pre-pandemic period. But during the pandemic, women who gave birth in the MAM model reported significant drops in their experience of respect ($p < 0.001$) and trust ($p < 0.001$), providing the lowest means of all three models.

To determine not just if the model was significant, but how much of an effect did it have on the outcome variance between the models, an Eta$^2$ was done. Analyzing the relationship between the models of care and outcomes, a large effect size was found during the pre-pandemic period for trust ($\eta^2$: 0·15), and in the pandemic period for respect ($\eta^2$: 0·23) and trust ($\eta^2$: 0·18). A larger effect size reflects a greater proportion of the variability in the outcome is accounted for by the models (Tables 5 & 6 for details).

**Table 3. Mean results during COVID by model.**

| Outcomes: | FEM- | MAM | NoM |
|---|---|---|---|
| | n = 363 | n = 312 | n = 515 |
| | (C19: 156) | (C19: 122) | (C19: 211) |
| **RESPECT** Covid mean (sd) | 58·0 (5·0) | 48·2 (6·63) | 52·6 (8·31) |
| **TRUST** Covid mean (sd) | 36·6 (2·71) | 33·2 (3·06) | 33·5 (8·89) |
| **FEAR** Covid mean (sd) | 7·1 (2·18) | 9·7 (2·99) | 7·6 (2·25) |
| **COVID KNOWLEDGE** Covid mean (sd) | 9·1 (3·5) | 8·7 (3·95) | 7·7 (3·69) |

FEM: Fully enabled midwifery; MAM: Midwives and medicine; NoM: No midwifery.

**Table 4. Difference in means from pre- to during COVID within models.**

| Outcomes: | FEM-<br>n = 363<br>(pre: 207, C19: 156) | MAM<br>n = 312<br>(pre: 190, C19: 122) | NoM<br>n = 515<br>(pre: 304, C19: 211) |
|---|---|---|---|
| **RESPECT** | +0·69 | -7·98 | -0·19 |
| Difference from pre- to during-COVID period. (CI) | (-0·30, 1·69) | (-9·32, -6·65) | (-1·66, 1·28) |
| p-value | p: 0·1713 | p: <0·001 | p: 0·7978 |
| **TRUST** | +0·11 | -1·78 | +0·31 |
| Difference from pre to during-COVID period (CI) | (-0·49,0·70) | (-2·44, -1·13) | (-0·38,0·98) |
| p-value | p: 0·7253 | p: <0·001 | p: 0·384 |

FEM: Fully enabled midwifery; MAM: Midwives and medicine; NoM: No midwifery.

**Table 5. Outcomes pre-COVID, with significance between the models.**

| | AOV p-value<br>(Were the models significantly related to the outcomes?) | Eta$^2$ ($\eta^2$) and [CI]<br>(How much effect did the models have on the outcome?) | TukeyHSD-pairwise comparison<br>p-value<br>Significance of difference between individual models | | |
|---|---|---|---|---|---|
| | | | MAM-FEM | NOM-FEM | NOM-MAM |
| **Respect Pre-Covid** | <0·001 | $\eta^2$ = 0·08 [0·05, 0·12] (medium effect) | 0·1526 | < 0·001 | < 0·001 |
| **Trust Pre-Covid** | < 0·001 | $\eta^2$ = 0·15 [0·12, 0·19] (large effect) | < 0·001 | < 0·001 | < 0·001 |

Effect Size: (eta squared is the proportion of the total variability in the dependent variable that is accounted for by the variation in the independent variable.)
Benchmarks: 0·01 = small, 0·06 = medium, 0·14 = large.

**Table 6. Outcomes during COVID, with significance between the models.**

| | AOV p-value<br>(Were the models significantly related to the outcomes?) | Eta$^2$ ($\eta^2$) and [CI]<br>(How much effect did the models have on the outcome?) | TukeyHSD- pairwise comparison<br>p-value<br>Significance of difference between individual models | | |
|---|---|---|---|---|---|
| | | | MAM-FEM | NOM-FEM | NOM-MAM |
| **Respect during Covid** | <0·001 | $\eta^2$ = 0·23 [0·18, 0·28] (large effect) | <0·001 | <0·001 | <0·001 |
| **Trust during Covid** | < 0·001 | $\eta^2$ = 0·18 [0·13, 0·23] (large effect) | < 0·001 | < 0·001 | 0·7904 |
| **Covid fear during Covid** | < 0·001 | $\eta^2$ = 0·16 [0·11, 0·20] (large effect) | < 0·001 | <0·09 | < 0·001 |
| **Covid Knowledge** | <0·05 | $\eta^2$ = 0·02 [0·00, 0·04] (small effect) | 0·6735 | <0·001 | 0·037 |

Effect Size: (eta squared is the proportion of the total variability in the dependent variable that is accounted for by the variation in the independent variable.)
Benchmarks: 0·01 = small, 0·06 = medium, 0·14 = large.

## COVID fear and knowledge

An ANOVA analysis found that the model of care a woman used for birth was significantly related to her experience of COVID-related fear and knowledge during COVID. The women cared for in the FEM model reported the lowest mean of COVID fear, when compared to the MAM models (<0.001) and the highest mean of COVID knowledge when compared to the NOM models (<0.001) (Table 6).

When analyzing the effect size of the models of care, a large effect size was found in the pandemic period for COVID-19 related fear ($\eta^2$: 0·16), suggesting the models of care accounted for the large variability in COVID-19 related fear (Table 6).

## Discussion

In this quasi-experimental study, a woman's positive experience of care in the fully enabled midwifery (FEM) model pre-COVID, was significantly higher than the midwives and medicine (MAM) model for trust, and significantly higher than the no midwives (NoM) model for both trust and respect. During COVID, women's positive experience of care in the FEM model was significantly higher than MAM for trust, respect and COVID fear, and significantly higher than NoM for trust and respect.

Midwives are being introduced globally in LMIC as one approach to address persistent high maternal mortality. The Lancet series on midwifery documented the potential impact of midwifery care [43], but cautioned the need for an enabling environment [44]. The use of midwifery-led continuity of care, which includes midwifery-led care units, is one approach being promoted [45].

The results of this study suggest the FEM model, a freestanding midwifery center, may serve as well or better than midwifery-led care units as an enabling environment for midwives in a low or middle income country. It also suggests the care by midwives in a freestanding midwifery center may provide more resilience during a crisis on women's experience of care than the MAM model–midwives working in midwifery-led care units in hospitals.

It's not surprising the midwifery centers did well. Midwives were working with other midwives, reinforcing and supporting the midwifery model of care, unlike other environments associated with burnout [46–48]. In addition, they were not directly affected by COVID as no COVID patients were treated in the freestanding midwifery centers. The separateness from the hospital system seemed to protect the women and midwives from exposure to, and stressors from, COVID. But it must be remembered that the full integration of midwifery centers within a health system is required to access the positive impacts, and health system resilience they may offer. Past research has shown the medical model fails to support professional midwives working to their full capacity [47–49], hence the global definition [36] and operational standards [38] for midwifery centers in LMIC can encourage development of consistent programs of care, high quality and an enabling environment for midwives [18].

Women who gave birth during the COVID-19 pandemic in the MAM model experienced a dramatic drop in their experiences, compared to the pre-pandemic period, reporting the lowest rates of respect and trust, and the highest rate of COVID related fear of *all three models*. Was there something unique about the three midwifery-led care units or the hospitals they were in, that made them more susceptible to stress from COVID? Could power dynamics between cadres impact behaviors, resulting in the poorer experiences of women? Is a midwifery-led care unit a good option for an enabling environment for midwives in LMIC? Considering most midwives work in hospitals alongside nurses, doctors and other health care workers, the results of the MAM model were surprising and require more exploration.

## Limitations

While this analysis involved a large sample across multiple sites, it does have limitations. A sub analysis of the outcomes by facility found some variation between facilities within all models they represented. There may have been differences such as management, supplies, support and staffing in addition to size and care provided, not just between the models, but from facility to facility.

For the outcome of respect, there were significant differences (p<0·001) between facilities for both the MAM and NoM models, and in terms of trust (p < 0·001), there were significant differences between facilities (p<0·001) within all of the models. For COVID fear, there were significant differences between facilities (p<0·001) in the NoM model. Additionally, the inter-facility variations seem larger between the MAM and NoM facilities. Although the variation between facilities is noted, the significance of differences between models on women's experiences and the optimal experiences of women in the FEM model was the objective of this study. (see S7 File for details on facility outcomes)

The phone survey, although allowing for anonymity, limited the participants to those who provided mobile numbers. Those without access to mobile phones may have had different experiences that were not captured.

## Conclusion

A woman's experience of respect and trust has been shown to drive demand for care [8, 50]. When that care is given by well-trained providers, with access to adequate supplies, maternal mortality has been shown to drop [1]. This study found that the model of care a woman gave birth in was strongly associated with her experience of respect and trust in the health care system. The highest reports of trust pre-pandemic, and trust and respect during the pandemic, and the lowest reports of COVID fear during the pandemic, were all found in the midwifery enabled environment of the freestanding midwifery centers.

Currently midwives are being promoted in many countries as a means of addressing maternal mortality. They are employed in hospitals and other facilities using the medical model of care. These findings suggest women served by midwives working in the medical model had poorer experiences of trust and respect compared to women served by midwives in freestanding midwifery centers.

Freestanding midwifery centers should be considered when promoting midwifery as a specific example of a HRBA for maternal and child health system strengthening in LMICs, as well as an enabling environment for midwives. Additionally, care provided by midwives working in midwifery centers may protect a woman's experience of care during a crisis, perhaps even more than midwives working in midwifery-led care units in traditional health care facilities, driving care seeking and offering much-needed resilience to the health system during a crisis.

This novel research looked specifically at freestanding midwifery centers as an enabling environment for the practice of midwifery. Due to the dearth of literature on this topic, this is the only study that compares this enabled midwifery model of care to traditional models in a LMIC during a crisis. Because midwifery centers provide a human rights-based care, are supportive of the midwifery model, are cost-efficient, and may offer protection during a crisis, further use of and research on this model is recommended.

## Supporting information

**S1 File. Ebola related fear and stigma questionnaire.**
(DOCX)

**S2 File. Women's survey.**
(DOCX)

**S3 File. Women participant response rate by model of care.**
(DOCX)

**S4 File. Facility data by model of care.**
(DOCX)

**S5 File. Description of participants in the different care models.**
(DOCX)

**S6 File. Group and covariates significance and coefficient.**
(DOCX)

**S7 File. Outcomes by facility.**
(DOCX)

**S8 File. Inclusivity in global research.**
(DOCX)

## Acknowledgments

The author would like to acknowledge in deep gratitude her research assistant, Noor Islam Pappu, data collectors, E Bithi Islam, Sumayea Tanzin Eti, Tahmina Tanha Khatan, and Fatema Tuj Johora, the generous involvement of Dr Selina Amin, and her staff at BRAC JPGSPH, and Kalpana Roy, for their time and commitment to this project. A special thank you to the midwives and staff at the midwifery centers in Jaintapur, Sylhet and Mirpur, Dhaka for their care, especially during the COVID crisis. Many thanks for all of your work and commitment to the women of Bangladesh.

## Author Contributions

**Conceptualization:** Jennifer Rebecca Stevens.

**Data curation:** Jennifer Rebecca Stevens.

**Formal analysis:** Jennifer Rebecca Stevens.

**Methodology:** Jennifer Rebecca Stevens.

**Project administration:** Jennifer Rebecca Stevens.

**Resources:** Jennifer Rebecca Stevens.

**Supervision:** Lora L. Sabin, Monica A. Onyango, Malabika Sarker, Eugene Declercq.

**Writing – original draft:** Jennifer Rebecca Stevens.

**Writing – review & editing:** Jennifer Rebecca Stevens, Lora L. Sabin, Monica A. Onyango, Malabika Sarker, Eugene Declercq.

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
