## [Decision Letter · Decision Letter 0]

8 Aug 2022

PONE-D-22-07498Midwifery centers as enabled midwifery:  Centering women’s experience of care with a human rights based approach before and during a crisisPLOS ONE

Dear Dr. Stevens,

Thank you for submitting your manuscript to PLOS ONE. After careful consideration, we feel that it has merit but does not fully meet PLOS ONE’s publication criteria as it currently stands. Therefore, we invite you to submit a revised version of the manuscript that addresses the points raised during the review process.

Please carefully consider constructive comments from both reviewers, particularly requests for clarification and suggestions for strengthening structure and clarity from reviewer #1.

We look forward to receiving your revised manuscript.

Kind regards,

Hannah Tappis, DrPH, MPH

Academic Editor

PLOS ONE

https://journals.plos.org/plosone/s/file?id=ba62/PLOSOne_formatting_sample_title_authors_affiliations.pdf".

Reviewers' comments:

Reviewer's Responses to Questions

**Comments to the Author**

1. Is the manuscript technically sound, and do the data support the conclusions?

Reviewer #1: Yes

Reviewer #2: Yes

2. Has the statistical analysis been performed appropriately and rigorously? 

Reviewer #1: Yes

Reviewer #2: Yes

3. Have the authors made all data underlying the findings in their manuscript fully available?

Reviewer #1: Yes

Reviewer #2: Yes

4. Is the manuscript presented in an intelligible fashion and written in standard English?

Reviewer #1: Yes

Reviewer #2: Yes

5. Review Comments to the Author

Reviewer #1: Dear authors, thanks for the opportunity for reviewing your manuscript on “Midwifery centers as enabled midwifery: Centering women’s experience of care with a human rights based approach before and during a crisis”. It is an interesting paper and fits the readers of PLOS ONE. I have a few comments to be considered before being published.

Title: a suggestion is to include the method used and in which country the study takes place in the title

Introduction

Overall, the introduction is well written, and fits well for a thesis, but it is far too long for a scientific paper. First the authors report from the Lancet maternal health series, WHO standards, Lancet series on midwifery, human rights approach, midwifery models of care and Covid, and Bangladesh, and it takes 6 pages before the readers come to the aim of the study.

As a reader I wonder, what the problem statement is of your study? If this is defined, it would help you to build your introduction in a shorter and more to the point. A suggestion is to shorten the introduction to be no longer than 1-2 pages.

Page 6, line 77-80: According to the ICM, by using the term midwife indicates that the person who has successfully completed a midwifery education programme that is based on the ICM Essential Competencies for Midwifery Practice ICM Definitions (internationalmidwives.org). Suggest that you refer to the persons not meeting the training standards of the International Confederation of Midwives for example nurse-midwives. This, as it will be less confusing when you talk about midwives on page 6 in relation to midwifery care.

Page 6-7, line, 103-107, the term fully enabled midwifery model (FEM), is this an accepted term? The reference which you use, talk about free standing midwifery unit

Page 7, line122, spell out NGO

Page 9, line 166, the aim mentions three models of care, but the introduction it talks only about one model. There needs to be a short sentence about that there are different models of care.

Method

Page 10, line 174-179: the three models; FEM, MAM and NOM, as a reader I am curious about these models, are they established in Bangladesh and are these terms and abbreviations being used there?

Page 10, line 183: How were the selected facilities invited?

For the reader to understand the different models, It is suggested to short describe the different care levels that exists in Bangladesh, and how these are being distributed in the provinces.

Page 11, line 190, add a reference related to midwives in Bangladesh which there are several articles published on.

Page 13, line 235, the phone-administered questionnaire consisting of 31 closed-ended, and 11 open-ended questions, was this a validated instrument being used, or what questionnaire is this?

Discussion

Page 19, Line349-388, Usually the discussion starts with a short summary of key findings, and thereafter being discussed again the literature. This is missing in this paper, and the discussion in its current form is partly a repetition of the results.

Please add a conclusion

Reviewer #2: Midwifery centers as enabled midwifery: Centering women’s experience of care with a

human rights based approach before and during a crisis is a well written manuscript. I suggest the last section on implementation could be divided into two with one heading named conclusion as it is very long and the reader must find the way in the text to find the conclusions in this section.

I suggest to correct type errors. The dot should come after every bracket with references, the heading to the left.

Good luck and I consider this minor revision.

6. PLOS authors have the option to publish the peer review history of their article (what does this mean?). If published, this will include your full peer review and any attached files.

Reviewer #1: No

Reviewer #2: No

---

## [Author Response · Author response to Decision Letter 0]

26 Oct 2022

Response to Reviewers

Dear Reviewers,

Thank you for taking the time to read my article and offer your consideration. I found the comments helpful and often reflected thoughts I had as well. Please find your comments below in black, with my response to each in blue.

Reviewer #1 comments: 

Title: a suggestion is to include the method used and in which country the study takes place in the title

Thank you, I agree. I have modified the title as suggested. 

Introduction.

Overall, the introduction is well written, and fits well for a thesis, but it is far too long for a scientific paper. First the authors report from the Lancet maternal health series, WHO standards, Lancet series on midwifery, human rights approach, midwifery models of care and Covid, and Bangladesh, and it takes 6 pages before the readers come to the aim of the study. As a reader I wonder, what the problem statement is of your study? If this is defined, it would help you to build your introduction in a shorter and more to the point. A suggestion is to shorten the introduction to be no longer than 1-2 pages.

I agree. I was concerned my reader would not understand the foundations of this research- a HRBA to care, the history of focusing on access to care, the shift that has occurred around quality of care and valuing a woman’s experience of care, and the unique models of midwifery care and midwifery centers. I have cut it back as much as I can and still present these clearly. 

Page 6, line 77-80: According to the ICM, by using the term midwife indicates that the person who has successfully completed a midwifery education programme that is based on the ICM Essential Competencies for Midwifery Practice ICM Definitions (internationalmidwives.org). Suggest that you refer to the persons not meeting the training standards of the International Confederation of Midwives for example nurse-midwives. This, as it will be less confusing when you talk about midwives on page 6 in relation to midwifery care. 

Thank you, this is a very important concept to be clear about. I tweaked the language to be as clear as possible. In many LMIC, midwives are called nurse-midwives as they are nurses who have had additional training to deliver babies. Many do not meet the ICM criteria. What we call them matters. I used the term “professional midwife” to identify them, not just “midwife” (a term that is used very often for health workforce cadres meeting a wide range of educational backgrounds, if they provide care during birth) and I chose the term “non-standardized midwife” for any cadre who does not meet the ICM standard. I hope this makes it as clear as possible, without offending anyone, yet clearly identifies that between the FEM and MAM models, I am comparing midwives who meet same ICM standard of education. (see line 71-74)

Page 6-7, line, 103-107, the term fully enabled midwifery model (FEM), is this an accepted term? The reference which you use, talk about free standing midwifery unit.

These models were currently in place throughout Bangladesh, but identified and named for this research. I made this clearer when they are first introduced and later in the design section. (see line 103-110, 162-164)

Page 7, line122, spell out NGO

Thank you, done. 

Page 9, line 166, the aim mentions three models of care, but the introduction it talks only about one model. There needs to be a short sentence about that there are different models of care.

This was added. (see line 97-110)

Method

Page 10, line 174-179: the three models; FEM, MAM and NOM, as a reader I am curious about these models, are they established in Bangladesh and are these terms and abbreviations being used there?

As mentioned above, this was expanded and explained in a few places. (see line 162-164)

Page 10, line 183: How were the selected facilities invited? 

This was added. (see line 168-171)

For the reader to understand the different models, It is suggested to short describe the different care levels that exists in Bangladesh, and how these are being distributed in the provinces.

I hate adding length to the paper, but I agree. This was added. (see line 135-141)

Page 11, line 190, add a reference related to midwives in Bangladesh which there are several articles published on.

Thank you, I agree. (see line 182)

Page 13, line 235, the phone-administered questionnaire consisting of 31 closed-ended, and 11 open-ended questions, was this a validated instrument being used, or what questionnaire is this?

The questionnaire was 3 surveys- 2 validated, one modified from an Ebola survey. This was specified in the article. (see line 225-236)

Discussion

Page 19, Line349-388, Usually the discussion starts with a short summary of key findings, and thereafter being discussed again the literature. This is missing in this paper, and the discussion in its current form is partly a repetition of the results. 

The discussion section was revised. (see line 337-380)

Please add a conclusion

Identified (see line 403-431).

Reviewer#2 Comments: 

I suggest the last section on implementation could be divided into two with one heading named conclusion as it is very long and the reader must find the way in the text to find the conclusions in this section.

Thank you. This section was revised with a clear conclusion identified at the end of the paper. 

I suggest to correct type errors. 

The dot should come after every bracket with references, the heading to the left.

Thank you. This was revised throughout the article.

---

## [Editor Report · Decision Letter 1]

15 Nov 2022

Midwifery centers as enabled environments for midwifery:   A quasi experimental design assessing women’s birth experiences in three models of care in Bangladesh, before and during covid.

PONE-D-22-07498R1

Dear Dr. Stevens,

We’re pleased to inform you that your manuscript has been judged scientifically suitable for publication and will be formally accepted for publication once it meets all outstanding technical requirements.

Kind regards,

Hannah Tappis, DrPH, MPH

Academic Editor

PLOS ONE
---

## [Editor Report · Acceptance letter]

23 Nov 2022

PONE-D-22-07498R1 

Midwifery centers as enabled environments for midwifery:   A quasi experimental design assessing women’s birth experiences in three models of care in Bangladesh, before and during covid. 

Dear Dr. Stevens:

I'm pleased to inform you that your manuscript has been deemed suitable for publication in PLOS ONE. Congratulations! Your manuscript is now with our production department. 

Kind regards, 

on behalf of

Dr. Hannah Tappis 

Academic Editor

PLOS ONE